# Improvement of In Vitro Seed Germination and Micropropagation of *Amomum tsao-ko* (Zingiberaceae Lindl.)

**Quyet V. Khuat** [1,2,*], **Elena A. Kalashnikova** [1], **Rima N. Kirakosyan** [1], **Hai T. Nguyen** [3], **Ekaterina N. Baranova** [4,5] and **Marat R. Khaliluev** [1,4,*]

1. Agronomy and Biotechnology Faculty, Russian State Agrarian University—Moscow Timiryazev Agricultural Academy, Timiryazevskaya 49, 127550 Moscow, Russia; kalash0407@mail.ru (E.A.K.); mia41291@mail.ru (R.N.K.)
2. Biology and Agricultural Engineering Faculty, Hanoi Pedagogical University 2, Nguyen Van Linh, Phuc Yen 15000, Vietnam
3. Biotechnology Faculty, Vietnam National University of Agriculture, Gia Lam, Hanoi 12406, Vietnam; nthaicnsh@vnua.edu.vn
4. All-Russia Research Institute of Agricultural Biotechnology, Timiryazevskaya 42, 127550 Moscow, Russia; greenpro2007@rambler.ru
5. N.V. Tsitsin Main Botanical Garden of Russian Academy of Sciences, Botanicheskaya 4, 127276 Moscow, Russia
* Correspondence: khuatvanquyet@hpu2.edu.vn or khuatquyetst@gmail.com (Q.V.K.); marat131084@rambler.ru (M.R.K.); Tel.: +7(965)173-96-65 (Q.V.K.); +7(499)-977-31-41 (M.R.K.)

**Abstract:** Black cardamom (*Amomum tsao-ko* Crevost & Lemarié) is a spice plant of great commercial value in Vietnam, but with limited propagation ability. Its seeds are characterized by a thick and hard seed coat, a small endosperm, and a small embryo, which are the causes of the physical dormancy of the seeds and low germination. Attempts in this study to improve the germination rate and achieve uniform germination included mechanical scarification, immersion in hot or cold water, acid scarification, and the application of plant growth regulators. Although immersion of seeds in cold water and application of plant growth regulators (PGRs) (gibberellic acid ($GA_3$) and 1-naphtaleneacetic acid (NAA)) showed positive effects on seed germination and subsequent seedling growth, mechanical scarification provided the highest germination rate of black cardamom seeds (68.0%) and significantly shortened germination time (53.7 days) compared to control (16.0% and 74.7 days). On the other hand, an efficient micropropagation protocol has been established using shoot tip explants derived from in-vitro-grown seedlings. Murashige and Skoog (MS) medium supplemented with 4.0 mg/L 6-benzylaminopurine (BAP) + 0.5 mg/L NAA proved to be most suitable for rapid multiplication and rooting, providing a mean of 5.4 shoots per explant, 6.8 cm shoot length, and 16.2 roots per explant after 7 weeks of culture. Well-rooted black cardamom plantlets have been successfully adapted to ex vitro conditions. "Fasco" bio-soil was more suitable for acclimatization, with a 48.9% survival rate, 23.3 cm plant length, and 5.7 leaves per plant after 3 months of planting. Improved germination and multiplication protocols can be used to improve propagation performances and to develop elite of black cardamom planting material.

**Keywords:** black cardamom; seed dormancy; mechanical scarification; shoot micropropagation

## 1. Introduction

*Amomum tsao-ko* Crevost & Lemarié, belonging to the ginger family (Zingiberaceae), is a valuable medicinal plant in Vietnam and is commonly known as "black cardamom", "do ho", or "sa nhan coc" [1,2]. It is a particularly shade-loving and moisture-loving species, so it can only be grown under the forest canopy, at an altitude of 1300–2200 m, with an average annual temperature of 13–15.3 °C, usually frequent fog, annual rainfall of 3500–3800 mm, and more than 90% humidity [1,3–5]. Therefore, it is distributed and cultivated in high

ranges with damp soil rich in humus and under the partial shade of evergreen forests belonging to the Northern Midland and Mountainous region of Vietnam (Lai Chau, Lao Cai, Yen Bai, Ha Giang, Tuyen Quang, and Cao Bang provinces). Additionally, black cardamom is also grown in China (Yunnan province) and Laos (Phongsaly province) [1,2,6]. In Vietnamese traditional medicine, black cardamom fruit is utilized to treat dyspepsia, nausea, malaria, bad breath, infections, etc. [1,7,8]. Black cardamom is also a spice used in many traditional Vietnamese dishes, e.g., Vietnamese beef noodle, chicken herbal dish, hotpot, etc.

For the above-mentioned reasons, black cardamom has become one of the main medicine crops for local people in the Northern Midland and Mountainous region of Vietnam. The area of black cardamom cultivation is constantly expanding. In Vietnam, local people use mainly seeds and rhizome segments to cultivate black cardamom. Because plants grown from rhizome segments are susceptible to diseases caused by viruses, fungi, or bacteria, they often provide a lower yield and fruit quality than plants grown from seeds. Additionally, the destructive harvesting of the rhizomes for vegetative propagation appears not to be workable because there is always the possibility of losing the mother plant during this process. However, the practice of seed propagation of local people illustrates that cardamom seeds germinate slowly and unevenly, and seedlings are slow growth. Studies on other species from the ginger family such as *Alpinia malaccensis* Roscoe [9], *Alpinia galanga* Willd. [10], korarima (*Aframomum corrorima* P.C.M. Jansen) [11,12], large cardamom (*Amomum subulatum* Roxb.) [13], and green cardamom (*Elettaria cardamomum* Maton) [14] have all shown that seed germination of these species was not fast and/or that many seeds do not germinate due to the presence of some kind of dormancy, possibly associated with the hard and impermeable nature of the seed coat. Additionally, in the above-mentioned articles, various seed treatments (including mechanical scarification, soaking in hot or cold water, acid scarification, and soaking in plant growth regulators) were applied to break seed dormancy. Presently, similar studies in black cardamom are lacking. Quyet et al. (2021) [15] found that surface disinfection with 0.1% $HgCl_2$ for 10 min showed the best seed disinfection effect, and MS medium diluted to 1/16 concentration was the best for in vitro germination of black cardamom seeds. Additionally, the authors applied a mechanical scarification treatment to improve the germination rate of dried seeds. However, the germination rate was reported to be quite low (33.3%). Determining the cause and applying more seed treatments to establish the best seed treatments in order to further improve the germination rate of these seeds is necessary.

Currently, the technique involving propagation using plant tissue promises to produce many disease-free and uniform-quality crops in a short time compared with traditional methods. In vitro propagation has been reported in various species of the genus *Amomum*, e.g., green cardamom [16–18], large cardamom [19–21], siam cardamom (*Amomum krevanh* Pierre) [22], white amomum (*Amomum villosum* Lour.) [23,24], purple amomum (*Amomum longiligulare* T.L. Wu) [25], and *Amomum* sp. [26]. Rhizome buds are used by most researchers as a source of explants when carrying out in vitro propagation of these species. Regarding black cardamom, Quyet et al. (2021) [27] carried out clonal micropropagation and also used rhizome buds as an explant source. In this study, the authors evaluated the separate effects of two cytokinins, 6-benzylaminopurine (BAP) and kinetin, in the fast multiplication phase and two auxins, indole-3-butyric acid (IBA) and 1-naphtaleneacetic acid (NAA), for root induction. Eyob S. (2009) [11], when carrying out the micropropagation of korarima, a species of the ginger family, used two different explant sources including shoot tip derived from in vitro grown seedlings and rhizome buds. The author found that the highest survival and number of shoots were obtained from the shoot tips of in vitro seedlings compared with rhizome buds. Reghunath (1989) [28] applied the shoot tips of green cardamom to perform micropropagation experiments through callus culture. The results of this study showed that Murashige and Skoog (MS) medium supplemented with the combination of 4.0 mg/L NAA or 1.0 mg/L 2,4-dichlorophenoxyacetic acid (2,4-D) and 1 mg/L BAP provided the highest callus induction rate after 28 days of culture. On the other hand, the combined effect of cytokines and auxins in micropropagation has been

reported to be effective in many species of the genus *Amomum*, including green cardamom, large cardamom, and purple amomum. These were not reported in the previous study [27]. Therefore, similar studies on black cardamom are needed to determine the most effective micropropagation protocol for black cardamom.

Therefore, the overall goal of the present study was to investigate the effects of different seed treatment methods on the seed germination and seedling growth of black cardamom under in vitro conditions and determine the effect of different PGRs on in vitro shoot multiplication and root formation of seedling shoot tips in order to develop a protocol that produces a constant supply of viable and clean clonally propagated plant materials for crop production.

## 2. Materials and Methods

### 2.1. Seed Material

Matured capsules of black cardamom were collected at the end of December 2020 from the cardamom forest in Tam Duong district, Lai Chau province, Vietnam, around 22°23′04.5″ N latitude and 103°32′44.0″ E longitude. After harvesting, they were rinsed with water, preserved by vacuum packing, and transferred to the biotechnology laboratory of the Russian State Agrarian University—Moscow Timiryazev Agricultural Academy. At the laboratory, seeds were extracted from the capsules, rinsed thoroughly with tap water to remove the aril, and used immediately for experiments (Figure 1).

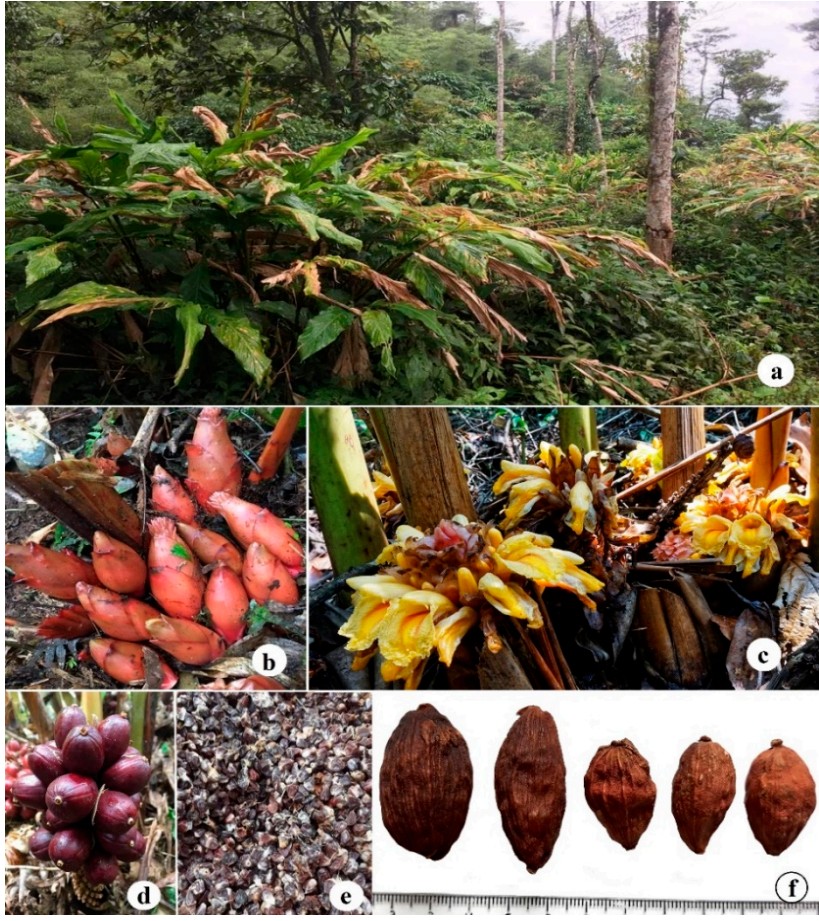

**Figure 1.** Black cardamom plants in Lai Chau province, Vietnam. (**a**) Black cardamom plants and their growing environment; (**b**) inflorescence shoots; (**c**) bloomed inflorescences; (**d**) fresh capsules; (**e**) fresh seeds; (**f**) dried ripe capsules (Photo by Ma A Chang, 2020).

A voucher specimen (No. QF 001) was deposited at the herbarium of Hanoi Pedagogical University N°2, Vietnam. Botanical identification was achieved by Dr. Tam H. M.

*2.2. Morphological and Anatomical Characteristics of Black Cardamom Seeds*

The characteristics of the shape, color, and size of seeds were examined via stereomicroscope "Stemi" DV4 (Carl Zeiss, Oberkochen, Germany). Detailed morphology and anatomy of seeds were evaluated using a scanning electron microscope (SEM). Black cardamom seeds were fixed in 2.5% glutaraldehyde in a 0.1 M Sorenson buffer, pH 7.2. After that, they were washed with buffer and dehydrated through ethanol series (30%, 50%, 70%, 96%, and 2 × 100%). In the next step, carbon dioxide ($CO_2$) for critical-point drying (Hitachi (Tokyo, Japan) HCP-2 critical point dryer) was applied. Dried seeds were mounted on an SEM stub with carbon-conductive tabs and coated with gold and palladium using an Eiko IB-3 ion coater (Eiko, Tokyo, Japan). Samples were observed and photographed under a JSM-6380LA SEM at 20 kV (JEOL, Tokyo, Japan).

*2.3. Experimental Design and Treatments*

Germination of black cardamom seeds is erratic, delayed, and poor under the traditional type of sowing. Attempts made to improve germination percentage and also to achieve uniform germination include mechanical scarification, immersion in hot or cold water, acid scarification, and soaking in PGRs (Table 1). Seeds without damage to the uniform size and color were selected randomly to conduct the experiments.

**Table 1.** Summary of black cardamom seed treatments.

| Treatment | Abbreviation | Description |
| --- | --- | --- |
| Mechanical scarification | ME | Soak in tap water for 24 h and scarify them manually by cutting 1–1.5 mm of the seed coat at the opposite site of hilum by a sterile scalpel. |
| Hot water | HW2m<br>HW4m | Soak in hot water at 100 °C for 2 min.<br>Soak in hot water at 100 °C for 4 min. |
| Cold water | CW | Soak in cold water at 4 ± 1 °C for 24 h. |
| Acid scarification | NAS10m<br>NAS15m<br>HAS10m<br>HAS15m | Soak in 50% nitric acid ($HNO_3$) for 10 min.<br>Soak in 50% nitric acid ($HNO_3$) for 15 min.<br>Soak in 25% hydrochloric acid (HCl) for 10 min<br>Soak in 25% hydrochloric acid (HCl) for and 15 min. |
| PGRs | GA$_3$24h<br>NAA24h | Soak in gibberellic acid (GA$_3$), 200 ppm, for 24 h.<br>Soak in 1-naphthylacetic acid (NAA) 200 ppm for 24 h. |

Control (without treatment). ME treatment are performed after disinfection seed step. Other treatments are performed before disinfection seed step. Each treatment was replicated three times with 25 seeds for each. To avoid contamination, only one seed was cultured per glass vessel.

*2.4. Obtaining of Aseptic Donor Seeds and Culture Conditions*

Above-mentioned treated seeds (Table 1) were rinsed with tap water. After that, seeds were washed in liquid soap for 10 min and then rinsed directly under running tap water. In the next step, seeds were disinfected in 70% ethanol for 30 s, followed by immersion in 0.1% (*w/v*) aqueous mercuric chloride for 10 min [15]. After surface disinfection, seeds were rinsed 4–5 times with sterile distilled water and transferred to the culture vessels containing basal MS medium [29] with macronutrients diluted to 1/16 strength for germination and growth [15]. The pH of the medium was adjusted to 5.6–5.8 by 1N NaOH before being autoclaved at 121 °C and 1.1 atm for 20 min. The cultures were maintained in a culture room at 25 ± 2 °C during a long-day photoperiod (16 h of light: 8 h of dark) with cool white fluorescent light (2000–2500 lux).

*2.5. Data Recording and Germination Assessment*

Seeds were considered germinated when the healthy, white radical had emerged through the integument. Data were scored from 30 to 90 days after culture. The contamination and germination seeds were counted every 10 days until the 90th day.

The following germination parameters were determined:

(1) Germination percentage (GP), the number of germinated seeds as a percentage of the total number of tested seeds, is given as:

$$GP = (\text{germinated seeds}/\text{total tested seeds}) \times 100\%;$$

(2) Mean germination time (MGT, days) is given according to Scott et al. [30] as:

$$MGT = \sum T_k N_k / S,$$

where $T_k$ is the number of days since the beginning of the experiment, $N_k$ the number of seeds germinated per day, and S is the total number of seeds germinated.

(3) Germination rate index (GRI) was calculated for each treatment using the following equation:

$$GRI = (G_1/1) + (G_2/2) + \ldots + (G_i/i),$$

where G is the germination day 1, 2, $\ldots$ , and i represents the corresponding day of germination [31].

Seedling length was measured using a ruler scale on the 110th day of culture.

### 2.6. Culture Media for Clonal Propagation

Seedlings (2–3 cm in length) were used as an explant source (shoot tips) for clonal propagation. They then were placed on agar-solidified MS medium supplemented with various concentrations of PGRs (auxin: 0.5 and 1.0 mg/L NAA, 1.0 mg/L and 2.0 mg/L 2,4-D); cytokinin: 1.0–4.0 mg/L BAP; see Table 3 for details). The experiment was arranged completely randomly and repeated three times, with 12 explants per treatment. Data were scored after 7 weeks. The length of shoots was measured using a ruler scale.

### 2.7. Plantlets Adaptation to Ex Vitro Conditions

After twelve weeks, in-vitro-propagated plantlets (4.0–5.0 cm in length, with 3–4 leaves) were acclimatized by transferring the culture vessels to the greenhouse for 5 days and then opening the cap of the culture vessels for 2 days. In the next step, they were taken out of the culture medium, and their basal portions were rinsed thoroughly in running tap water to remove agar and traces of medium. After that, these plantlets (n = 30 for each variant) were treated with 0.5% (*w/v*) Bavistin solution (10 min) to prevent fungal contamination and planted directly on 2 different types of soil including "Garden Star" soil universal (consisting of peat and mineral fertilizers) and "Fasco" bio-soil (consisting of high- and lowland peat, sand, bio-humus, dolomite flour, and complete mineral fertilizer). Survival rate of the plantlets and their morphometric characteristics were recorded after 1 and 3 months of soil adaptation.

### 2.8. Statistical Analysis

Mean values of all data were calculated using Microsoft Office Excel 2013 packages. Analysis of variance (ANOVA) was performed using Statistica, version 10.0, and means were compared using Duncan's multiple range test at a significance level of $\alpha = 0.05$.

## 3. Results

### 3.1. Morphological and Anatomical Characteristics of Black Cardamom Seeds

Black cardamom seeds are conical, polyhedral, 0.35–0.7 cm in diameter, brown or black, and covered with greyish-white membranous aril (Figure 2a–c). They have a pungent, slightly spicy, and aromatic taste. The weights of 100 fresh and dried seeds were found as 9.5 and 6.1 g, respectively.

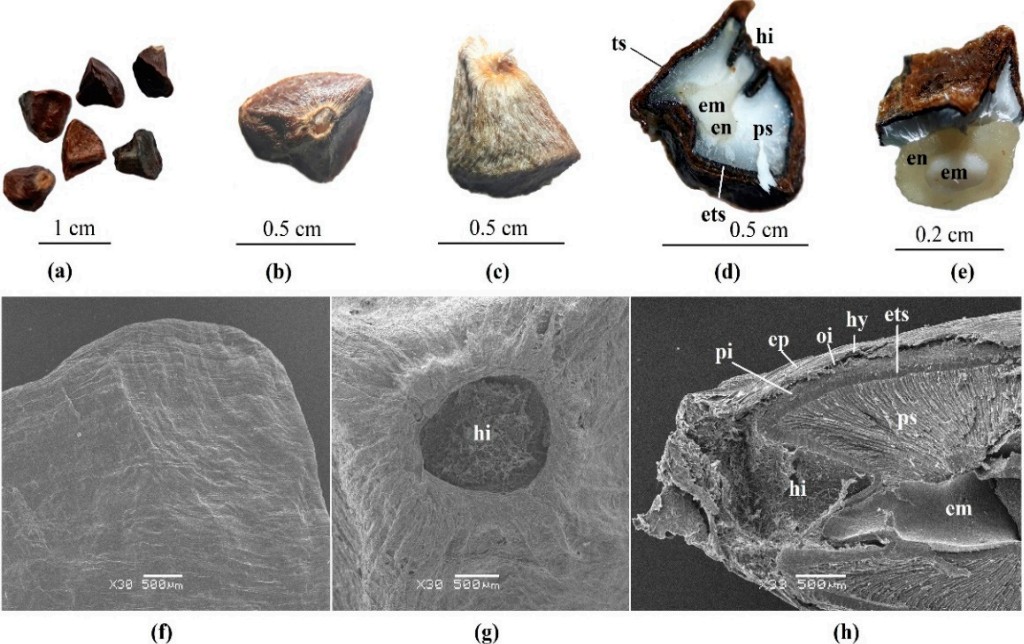

**Figure 2.** Black cardamom seeds: (**a**,**b**) morphological characteristics of seeds; (**c**) seed covered by arillus; (**d**,**e**) anatomical characteristics of seeds under the stereomicroscope: ts—testa, ets—endotesta, hi—hilum, ps—perisperm, en—endosperm, em—embryo; (**f**–**h**) anatomical characteristics of seeds under the SEM: ep—epidermal cells of testa, hy—hypodermis, oi—oil cell layer, pi—pigment layer. Scale bars (**f**–**h**) = 500 μm.

The longitudinal section of the black cardamom seed was observed by stereo microscope and SEM (Figure 2d–h). From the outside to the inside of the seed, the observed structures include:

(1)　Testa (or seed coat): very hard, representing an effective barrier against water and air, and consisting of: epidermis, comprising one layer of cells, longitudinally elongated; hypodermis, comprising one layer of cells, tangentially elongated; below the hypodermis, a layer of large parenchymatous cells containing volatile oil; below this layer, the pigment layer comprising several layers of brown cells; endotesta, comprising one layer of palisade sclerenchymatous cells, brown.
(2)　Perisperm: well-developed, composed of parenchymatous cells, white, and encircling the endosperm and embryo.
(3)　Endosperm: small, greyish-white, and partially enclosing the embryo.
(4)　Embryo: linear and fully developed.

### 3.2. In Vitro Germination Steps of Untreated Black Cardamom Seeds

In the in vitro germination of untreated black cardamom seed (control), the radicle emerges from the hilum of the seed and forms the primary root at around the 75th day of culture (Figure 3a). After that, the coleoptile emerges, grows to approximately 0.2–0.3 cm, then stops growing and will be pierced by the first leaf of the plumule (Figure 3c,d). The first leaf will grow, expand, and be supplemented by additional leaves (Figure 3e). At the other end of the embryonic axis, the primary root soon dies while adventitious roots (roots that arise directly from the shoot system) emerge, develop, and will ultimately produce a fibrous root system (Figure 3b).

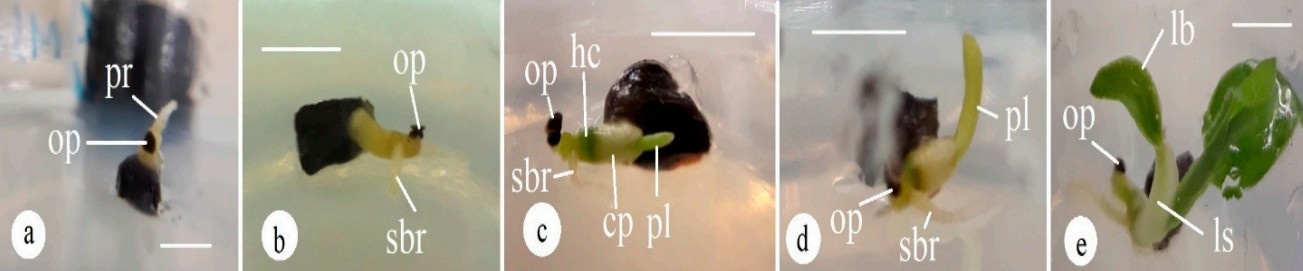

**Figure 3.** In vitro germination steps of untreated black cardamom seed: (**a**,**b**) radicle emerges (at 75th day of culture): op—operculum, pr—primary root, sbr—shoot-borne root; (**c**) coleoptile emerges and is pierced by the primary leaf (at 85th day of culture): cp—coleoptile, hc—hypocotyl, pl—primary leaf; (**d**,**e**) the first leaf grows, expands, and is supplemented by additional leaves (at 85th and 110th day of culture, respectively): lb—leaf blade, ls—leaf sheath. Scale bars = 0.5 cm.

*3.3. Effect of Seed Treatments on the Germination and Seedling Growth of Black Cardamom under In Vitro Conditions*

Efficiency of seed surface disinfection by 0.1% $HgCl_2$ for 10 min was significantly different between seed treatments after 10 days of culture (Table 2).

**Table 2.** Different seed treatment effects on seed germination and seedling growth of black cardamom cultured in vitro.

| Treatment | Contamination Free (%) | GP [2] (%) | MGT [3] (days) | GRI [4] | Seedling Length (cm) | No. of Leaves |
|---|---|---|---|---|---|---|
| Control | 68.0 ± 2.3 [1] c | 16.0 ± 2.1 c | 74.7 ± 2.9 b | 0.05 ± 0.01 e | 2.09 ± 0.11 bc | 1.83 ± 0.15 b |
| ME | 69.3 ± 2.7 c | 68.0 ± 4.0 a | 53.7 ± 0.7 d | 0.34 ± 0.02 a | 3.05 ± 0.07 a | 3.07 ± 0.02 a |
| HW2m | 66.7 ± 2.7 c | 8.0 ± 2.3 d | 85.0 ± 2.9 a | 0.02 ± 0.01 f | 1.87 ± 0.08 cd | 1.62 ± 0.25 b |
| HW4m | 68.0 ± 4.6 c | 5.3 ± 1.0 d | 88.3 ± 1.7 a | 0.02 ± 0.00 f | 1.66 ± 0.08 d | 1.58 ± 0.28 b |
| CW | 68.0 ± 2.3 c | 28.0 ± 2.2 b | 58.9 ± 1.4 cd | 0.13 ± 0.01 b | 2.25 ± 0.05 b | 2.70 ± 0.1 a |
| NAS10m | 78.7 ± 1.3 b | 16.0 ± 2.0 c | 61.4 ± 3.3 c | 0.07 ± 0.01 de | 2.17 ± 0.07 bc | 1.75 ± 0.14 b |
| NAS15m | 84.0 ± 2.3 ab | 22.7 ± 1.3 bc | 61.1 ± 2.0 c | 0.10 ± 0.01 cd | 2.23 ± 0.1 b | 1.80 ± 0.17 b |
| HAS10m | 82.7 ± 1.3 ab | 22.7 ± 1.3 bc | 62.3 ± 1.5 c | 0.10 ± 0.01 cd | 2.10 ± 0.17 bc | 2.62 ± 0.09 a |
| HAS15m | 86.7 ± 1.3 a | 17.3 ± 1.0 c | 62.5 ± 3.8 c | 0.08 ± 0.01 de | 2.07 ± 0.11 bc | 2.58 ± 0.14 a |
| GA$_3$24h | 66.7 ± 4.8 c | 29.3 ± 1.2 b | 61.4 ± 0.8 c | 0.13 ± 0.01 b | 2.93 ± 0.1 a | 2.79 ± 0.16 a |
| NAA24h | 62.7 ± 2.7 c | 25.3 ± 1.1 b | 60.6 ± 2.2 cd | 0.11 ± 0.01 bc | 2.81 ± 0.16 a | 2.95 ± 0.07 a |
| LSD$_{0.05}$ | 5.3 | 4.7 | 6.8 | 0.02 | 0.31 | 0.47 |

Treatments: control; ME—soak in water for 24 h + scarify by scalpel; HW2m and HW4m—soak in hot water at 100 °C for 2 and 4 min, respectively; CW—soak in cold water for 24 h; NAS10m and NAS15m—soak in 50% $HNO_3$ for 10 and 15 min, respectively; HAS10m and HAS15m—soak in 25% HCl for 10 and 15 min, respectively; GA$_3$24h—soak in 200 ppm GA$_3$ for 24 h; NAA24h—soak in 200 ppm NAA for 24 h. Means followed by a different letter are significantly different at an alpha level of 0.05 according to the Duncan's multiple range test. Percentage values were arcsin $\sqrt{X}$ transformed prior to statistical analysis. [1] Mean ± standard error. [2] GP (%) = germination percentage. [3] MGT (days) = mean germination time. [4] GRI = germination rate index.

Acid seed scarification had a significantly higher percentage of contamination-free seed compared with other treatments. Among these, seeds were dipped into 25% HCl for 15 min and disinfected with 0.1% $HgCl_2$ for 10 min in the next step and gave the highest percentage of contamination-free seeds, reaching 86.7%. In the other seed treatments, there was not a statistically significant difference in the percentage of contamination free seed compared to the control.

Results also showed that there was a significant difference in germination of black cardamom seeds under different seed treatments at α = 0.05 (Table 2). Applying mechanical scarification (ME) provided the highest mean germination percentage (reaching 68.0%) and the lowest mean germination time (reaching 53.7 days) after 90 days of culture. In this treatment, it took only approximately 50 days for 50% seed germination, while all other treatments had a lower germination percentage of 50% (Figure 4). Treatment of seed

immersion in cold water (CW) or PGRs (GA$_3$24h and NAA24h) also showed significantly higher mean germination percentages than the control. Acid seed scarification methods (NAS10m, NAS15m, HAS10m, and HAS15m) did not show a significant effect on the germination percentage of black cardamom seeds, but they significantly reduced the seed germination time compared to the control. Hot-water-soaking treatments (HW2m and HW4m) appeared to have a negative effect on seed germination (Table 2). In addition, after the 90th day of culture, seed germination was not observed in any of the treatments.

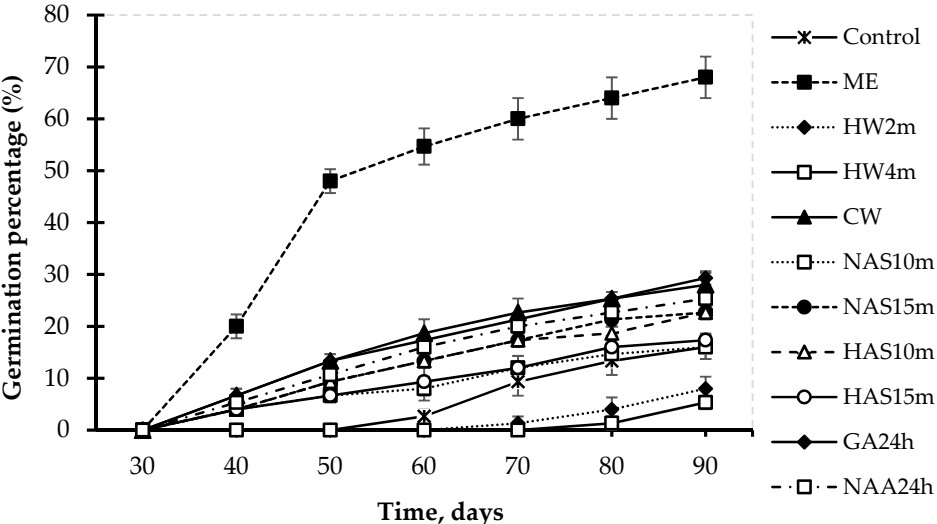

**Figure 4.** In vitro black cardamom seed germination patterns under different treatments.

Generally, the ME, GA$_3$24h, and NAA24h treatments significantly improved in vitro seedling length and number of leaves after 20 days of subsequent culture from the 90th day. In the other seed treatments, there were not statistically significant differences in mean seedling length and mean number of leaves compared to the control (Table 2, Figure 5).

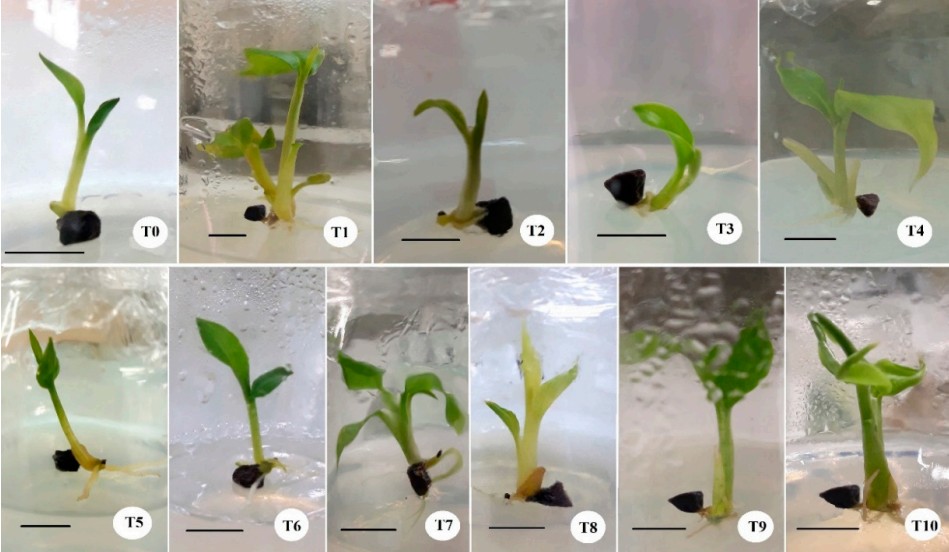

**Figure 5.** Seedling growth of black cardamom at the 110th day under different seed treatments: T0—Control; T1—ME (soak in water for 24 h + scarify by scalpel); T2—HW2m (soak in hot water at 100 °C for 2 min); T3—HW4m (soak in hot water at 100 °C for 4 min); T4—CW (soak in cold water for 24 h); T5—NAS10m (soak in 50% HNO$_3$ for 10 min); T6—NAS15m (soak in 50% HNO$_3$ for 15 min); T7—HAS10m (soak in 25% HCl for 10 min); T8—HAS15m (soak in 25% HCl for 15 min); T9—GA$_3$24h (soak in 200 ppm GA$_3$ for 24 h); T10—NAA24h (soak in 200 ppm NAA for 24 h). Scale bars = 1 cm.

### 3.4. Effect of Different PGRs on Shoot Multiplication, Elongation, and Rooting in Black Cardamom

After 7 weeks of culture on the MS medium supplemented with various concentrations of PGRs (auxin and cytokinin), the growth responses of the explants (shoot tip explants derived from in vitro grown seedlings) were different (Table 3).

**Table 3.** Effect of different PGRs in MS culture medium on shoot multiplication, elongation, and rooting in black cardamom.

| Type and Concentration of PGRs (mg/L) | | | Shoots Number (units) | Shoot Length (cm) | Root Number (units) | Callus Induction (%) | Shoot and Root Quality |
|---|---|---|---|---|---|---|---|
| **BAP** | **NAA** | **2,4-D** | | | | | |
| 0.0 | 0.0 | 0.0 | $0.6 \pm 0.1$ d | $3.0 \pm 0.1$ d | $2.7 \pm 0.2$ d | 0.0 e | + |
| 1.0 | 0.0 | 0.0 | $3.4 \pm 0.3$ c | $5.1 \pm 0.1$ c | $5.6 \pm 0.2$ c | 0.0 e | ++ |
| 1.0 | 4.0 | 0.0 | 0.0 e | 0.0 e | 0.0 e | 0.0 e | |
| 1.0 | 0.0 | 1.0 | 0.0 e | 0.0 e | 0.0 e | $83.3 \pm 5.3$ a | |
| 1.0 | 0.0 | 2.0 | 0.0 e | 0.0 e | 0.0 e | $58.3 \pm 8.3$ b | |
| 2.0 | 0.5 | 0.0 | $3.7 \pm 0.5$ c | $5.1 \pm 0.1$ c | $5.4 \pm 0.3$ c | 0.0 e | ++ |
| 2.0 | 1.0 | 0.0 | $3.8 \pm 0.4$ c | $5.5 \pm 0.2$ b | $5.8 \pm 0.2$ c | 0.0 e | ++ |
| 3.0 | 0.5 | 0.0 | $4.0 \pm 0.4$ bc | $5.8 \pm 0.2$ b | $5.9 \pm 0.5$ c | 0.0 e | ++ |
| 3.0 | 1.0 | 0.0 | $4.1 \pm 0.3$ bc | $5.8 \pm 0.1$ b | $6.3 \pm 0.3$ c | 0.0 e | +++ |
| 4.0 | 0.5 | 0.0 | $5.4 \pm 0.3$ a | $6.8 \pm 0.3$ a | $16.2 \pm 0.8$ a | 0.0 e | +++ |
| 4.0 | 1.0 | 0.0 | $4.9 \pm 0.2$ ab | $5.8 \pm 0.2$ b | $8.9 \pm 0.7$ b | 0.0 e | ++ |
| | LSD$_{0.05}$ | | 0.9 | 0.5 | 1.2 | 9.5 | |

Means followed by the same letter are not significantly different at $\alpha = 0.05$ according to the Duncan's multiple range test. Values of callus induction were arcsin $\sqrt{X}$ transformed prior to statistical analysis. The data have been recorded on a per-explant basis and recorded after 7 weeks of culture. +: small shoots, thin and light green leaves, and thin and few root-hairs roots; ++: medium shoots, thin and light green leaves, and medium and a few root-hairs roots; +++: fat shoots, thick and dark green leaves, and fat and many root-hairs roots.

Results indicated that the combination of BAP and NAA showed a high response for both shoot, leaf, and root induction after seven weeks of culture. MS medium supplemented with 4.0 mg/L BAP and 0.5 mg/L NAA exhibited the overall best response (mean shoot number: $5.4 \pm 0.3$, mean shoot length: $6.8 \pm 0.3$ cm, and mean root number: $16.2 \pm 0.8$) (Table 3). Shoots developed in clusters, stout pseudo-stems, thick and dark green leaves, and fat and many root-hairs roots (Figure 6c,e–g). The second-highest response was obtained using MS culture medium supplemented with 4.0 mg/L BAP and 1.0 mg/L NAA (mean shoot number: $4.92 \pm 0.22$, mean shoot length: $5.8 \pm 0.2$ cm, and mean root number: $8.9 \pm 0.7$) (Table 3). The in vitro response of seedling shoot tips on medium without PGRs (control) was low (mean shoot number: $0.6 \pm 0.1$, mean shoot length: $3.0 \pm 0.1$ cm, and mean root number: $2.7 \pm 0.2$) (Table 3). It can be seen that MS medium supplemented with a combination of BAP at high concentration and NAA is very suitable not only for shoot multiplication but also for root formation, and, hence, a separate constitution for root initiation is not required, as is often the case.

On the other hand, the combination of high-concentration auxin (including NAA and 2,4-D) with lower-concentration cytokinin (BAP) also showed different effects on callus induction of explants. The treatments related to 2,4-D produced a high callus induction rate (Table 3). However, produced callus were white and did not initiate shoot organogenesis in any of the explants when they were subjected to callogenesis induction treatments (Figure 6d). At the combination of 4.0 mg/L NAA with 1.0 mg/L BAP, callus induction was not observed; the explants were increasingly dark brown to blackish and, finally, rotten.

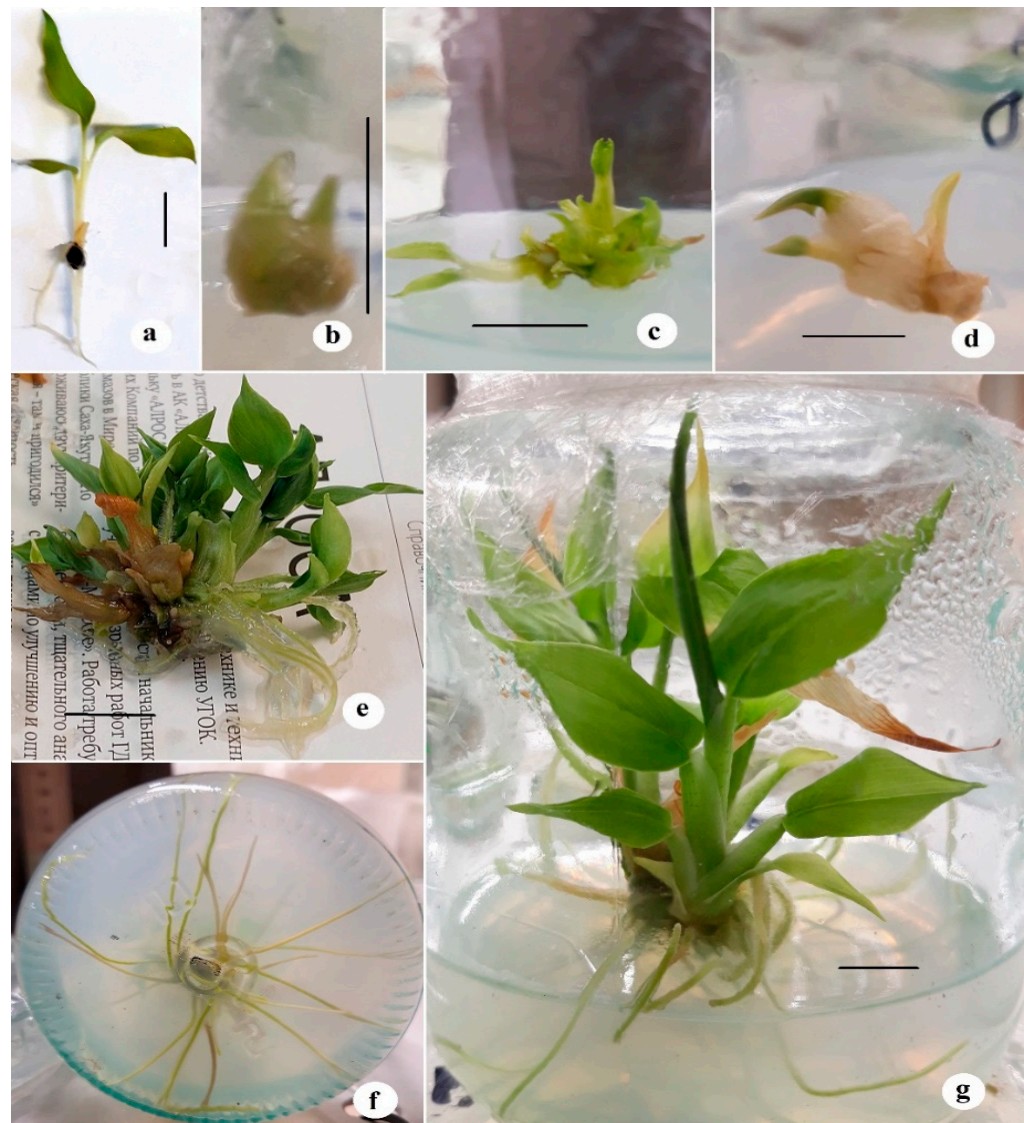

**Figure 6.** Different stages of in vitro micropropagation of black cardamom through seedling shoot tip culture on MS medium supplemented with BAP (4 mg/L) and NAA (0.5 mg/L) (except Figure (**d**)): (**a**) in vitro seedling, the source of explant; (**b**) explant (seedling shoot tip) after 1 week; (**c**,**e**,**g**) cluster of multiple shoots after 3, 5, and 7 weeks; (**f**) rooting after 7 weeks; (**d**) callus induction on MS medium supplemented with BAP (1 mg/L) and 2,4-D (1 mg/L) after 7 weeks. Scale bars = 1 cm.

*3.5. Plantlets Adaptation to Ex Vitro Conditions*

After being acclimatized to changes in temperature, humidity, and water loss, the in-vitro-propagated plantlets (4.0–5.0 cm in length, with 3–4 leaves (Figure 7a)) were directly planted on two soil types. The results presented in Table 4 show that the mean survival rate of in vitro plantlets on the two soil types reached 75.6–83.3% after a month of planting and 41.1–48.9% after 3 months of planting (Figure 7b,d). Among the two soil types studied, "Fasco" bio-soil produced a higher survival rate of 83.3% after a month of planting and 48.9% after 3 months of planting. The plantlets grew on this soil type better (mean length of 23.3 cm and a mean number of leaves of 5.7). The corresponding results when planting on "Garden Star" soil were all lower (Figure 7c,e).

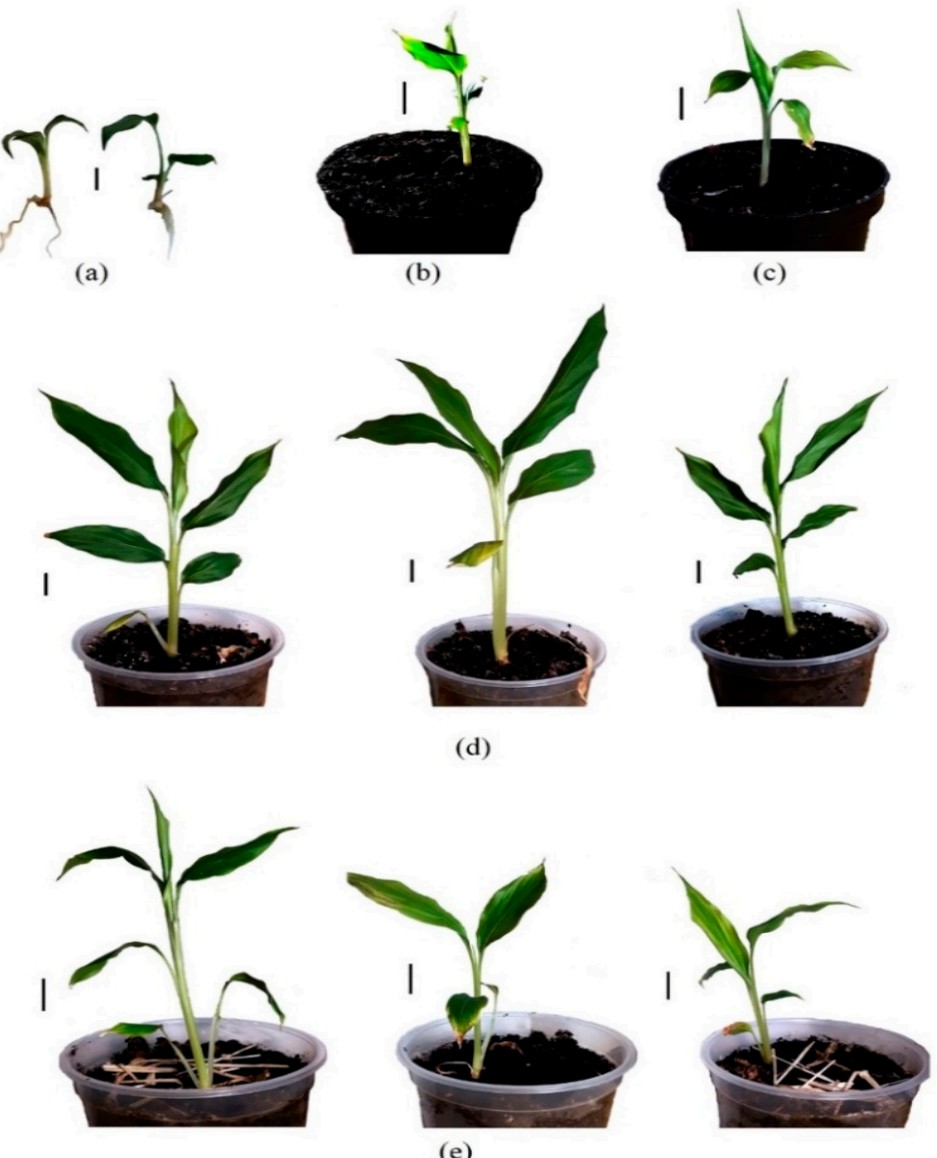

**Figure 7.** Effect of acclimatization of in-vitro-propagated black cardamom plantlets after transfer in the greenhouse conditions: (**a**) in-vitro-propagated plantlets; plantlets after 1 (**b**) and 3 (**d**) months of planting on "Fasco" bio-soil; plantlets after 1 (**c**) and 3 (**e**) months of planting on "Garden Star" soil. Scale bar = 2 cm.

**Table 4.** Effect of different soil types on efficiency of black cardamom plantlet adaptation and their morphometric characteristics.

| Soil Type | Survival Rate (%) | | Plantlet Length (cm) after 3 Months | No. of Leaves after 3 Months |
|---|---|---|---|---|
| | After a Month | After 3 Months | | |
| Garden Star | 75.6 ± 4.8 [1] | 41.1 ± 4.2 | 13.2 ± 0.6 | 4.3 ± 0.2 |
| Fasco | 83.3 ± 1.9 | 48.9 ± 4.8 | 23.3 ± 0.5 | 5.7 ± 0.2 |

[1] Mean ± standard error.

## 4. Discussion

The objectives of the study reported in this paper were to find solutions to improve the germination rate of black cardamom seeds, thereby creating an explant source for micropropagation to create many disease-free crops with outstanding characteristics of the selected mother plant to provide for mass production.

Seed coat hardness is an important factor that affects germination in seeds [32]. Seed dormancy has been reported in several species of the Zingiberaceae, including *Alpinia malaccensis* Roscoe [9], *Alpinia galanga* Willd. [10], korarima [11,12], large cardamom [13], and green cardamom [14]. In black cardamom, the morphological and anatomical characteristics of seeds established in the study, including the very hard testa, small endosperm (i.e., possessing low food storage), and small embryo, can be responsible for the difficulty of seed germination and the slow growth of seedlings. The strong inhibitory effect of the seed coat on seed germination may be caused by several possible mechanisms, including mechanical constraint, prevention of water and oxygen uptake, and retention or production of chemical inhibitors [33–35]. The integument breaking or softening, for instance, is needed to remove dormancy imposed by seed coat hardness or impermeability. Several authors (Alamgir and Hossain 2005 [36], 2005 [37]; Azad et al., 2006 [38], 2006 [39], 2010 [40], 2010 [41], 2011 [42]) have discussed different methods of pre-sowing treatments for seed germination in order to break dormancy, enhance the rate of germination, and speed up the germination process. Our studies showed that the seed treatments had a significant effect on the germination and growth parameters of black cardamom seeds. Among the applied seed treatments, mechanical scarification produced the best effect on seed germination parameters and subsequent seedling development. The findings of our previous study [15] also displayed similar results, but the germination rate of seeds was significantly improved in this experiment. The difference in the type of seeds used and the time of sowing in our two experiments is probably the cause for that difference. While in the first report we used dried seeds that were collected in October and dried in the open sun for 3 days, in this experiment we used fresh seeds that were collected in December. According to local people's experience, the best time to sow black cardamom seeds is at the end of December, and they must be sown immediately after harvest because they will lose their ability to germinate when stored for a long time. In a study on the effect of moisture in seeds on the germination of green cardamom seeds, Sangakkara U. R. (1990) [43] found that reducing the moisture content of seeds significantly reduced and even destroyed the viability of them. Robert (1973) [44] termed them recalcitrant seeds. Bearing similar physiological characteristics to green cardamom seeds may be responsible for the difference in results in our two experiments. Further studies are needed to confirm this issue. Copeland (1995) [45], Hartmann (1997) [46], Missanjo E. (2014) [47], and Botseleng B. (2014) [48] also confirmed that mechanical scarification produced the best effect in breaking the dormancy of seeds. However, recent studies on breaking dormancy of some Zingiberaceous species have shown that the effect of chemical scarification treatments is better than mechanical scarification treatments. According to Radhamani et al. [49], the treatments of green cardamom seeds in India with 25% sulfuric acid ($H_2SO_4$) for ten minutes and 80% absolute alcohol for 30 min were the most effective treatments in breaking seed dormancy. Dahanayake [14] reported that soaking in 50% nitric acid ($HNO_3$) for 15 min was the most effective for green cardamom seeds' dormancy in Sri Lanka. In our study, acid scarification treatments were not effective in increasing the germination rate of black cardamom seeds. Similarly, the report of Seid et al. [50] also showed that acid scarification treatments were not effective in breaking dormancy of green cardamom seeds. According to this report, soaking in 80% alcohol for 30 min was the most effective method for breaking the dormancy of green cardamom seed. Endogenous gibberellins have been widely studied in relation to the breaking of seed dormancy in various species. $GA_3$ has been exogenously applied as a substitute for stratification and has increased germination in many plant species, e.g., *Fagus sylvatica* L. [51], *Crataegus pseudoheterophylla* Pojark. [52], and *Juniperus polycarpos* K. Koch [53]. In a previous study on korarima, combined treatment of 50% $H_2SO_4$ for 60 min and 250 mg/L $GA_3$ for 24 h yielded the highest dormancy-breaking effect [11]. In this study, the positive effects of PGRs on the seed germination efficiency and growth of black cardamom seedlings were also noted. Hot water treatments have been reported to enhance germination of hard-coated seeds by elevating water and $O_2$ permeability of the testa [54]. Rivai et al. [9] showed that treatment with hot water at 75 °C for 5 min gave the highest germination rate of *Alpinia malaccensis* Roscoe seeds. However, in our study, the hot water

treatment seemed to have a negative effect on the seed germination and growth of black cardamom seedlings. This result is similar to that of Okunlola et al. (2010) [55] regarding African locust bean (*Parkia biglobosa* Benth.). Cold water treatment of black cardamom gave a fair germination percentage and a reduced mean germination time when compared to hot water and control treatments. This also implies that soaking in cold water could also reduce the dormancy period in seeds of black cardamom when compared to control. This result concurs with earlier reports of Emerhi and Nwiisuator (2010) [56] and Falemara et al. (2014) [57] that show that soaking in cold water is a feature that enhances germination in seeds of tropical trees. The subsequent increase in the germination percentage, decrease in mean germination time, and increase in germination index when subjected to different treatment methods are indications that the hard seed coat is responsible for the dormancy in black cardamom.

The next step in our study was to apply shoot tip explants derived from in-vitro-grown seedlings to establish an efficient in vitro propagation procedure that could be used for commercial purposes. In a previous study on large cardamom [19], when rhizome segments were cultured on an MS medium supplemented with BAP or NAA alone or in a combination of BAP + IBA, BAP + NAA, or IBA + NAA, they produced shoots and roots simultaneously. These researchers reported 90% survival of plants after transfer to soil [19]. In the present investigation, the MS medium supplemented with 4.0 mg/L BAP and 0.5 mg/L NAA was found to be suitable for large-scale multiplication of black cardamom. The use of MS medium supplemented with BAP at high concentration and NAA significantly shortened the time of in vitro plantlet generation (7 weeks) compared with our previous study (14 weeks). Earlier reports on the clonal micropropagation of other Zingiberaceous species such as green cardamom, ginger, and turmeric, indicated similar results [16,58–62]. According our previous study [27], MS medium supplemented with 1 mg/L BAP without auxin was best for shoot multiplication of rhizome buds of black cardamom after six weeks of culture (mean shoot number and mean shoot length were 4.5 and 5.5 cm, respectively). However, in this study, the growth response of shoot tip explants derived from in vitro grown seedlings to this medium was moderate (mean shoot number: $3.4 \pm 0.3$, mean shoot length: $5.1 \pm 0.1$ cm, and mean root number: $5.6 \pm 0.2$) (Table 3).

"Fasco" and "Garden Star" are two commonly used soils in the Russian Federation because they have a composition suitable for most crops grown here. The main components of these two soils are peat and mineral fertilizers. However, in the composition of "Fasco" bio-soil, there is also bio-humus—an organic fertilizer containing many beneficial microorganisms for the soil, which helps to increase the fertility and aeration of the soil. This difference may explain the more positive effect of "Fasco" bio-soil on survival and subsequent plantlets growth compared with "Garden Star" soil.

## 5. Conclusions

This study is one of the first reports of the morphological, anatomical, and germinating characteristics of black cardamom seeds collected in Vietnam. Seed coat ultrastructural features of the black cardamom using SEM were first reported. Because of the consistency in seed morphological features, SEM is considered useful when it comes to phylogenetic information and a great potential source of taxonomy. Additionally, we identified black cardamom seeds as physical dormancy seeds and the hard seed coat as the major cause. Applying mechanical scarification treatment before sowing has been shown to be the most effective for improving seed germination rates. Methods of immersion in cold water or PGRs before sowing are also recommended, although the effect of dormancy breaking is not as high as that of the mechanical scarification method. Traditional methods of propagation of black cardamom (by seeds and rhizomes) do not currently meet the increasing demand for large-scale production of crops. Therefore, the developed in vitro propagation protocol can be an effective solution for rapid multiplication of high-yielding elite plants to meet the needs of expanding cultivation of this important crop, but it needs to be tested for large-

scale multiplication and field planting. It can be said that this investigation carries great commercial significance, as black cardamom is used as a spice in many Asian countries, as a flavoring agent, and for pharmaceutical purposes by other industries.

**Author Contributions:** Q.V.K. and R.N.K.—experiments on black cardamom culture in vitro. E.N.B.—preparation of plant material for SEM; Q.V.K. and H.T.N.—statistical analysis of experimental data; E.A.K.—manuscript conceptualization. Q.V.K., E.A.K. and M.R.K.—writing of the manuscript; Q.V.K. and M.R.K.—reviewing and editing of the manuscript; all authors approved the final manuscript for publication and agreed to be accountable for all aspects of the manuscript. All authors have read and agreed to the published version of the manuscript.

**Funding:** The results of Sections 3.1–3.3 were supported by assignments 122020300187-2 (Tsitsin Main Botanical Garden of Russian Academy of Sciences) and 0431-2022-0003 (All-Russia Research Institute of Agricultural Biotechnology) of the Ministry of Science and Higher Education of the Russian Federation, respectively. Additionally, Sections 3.4 and 3.5 were supported by the Ministry of Science and Higher Education of the Russian Federation in accordance with agreement № 075-15-2022-746, dated 13 May 2022 (internal number MK-3084.2022.1.4) on providing a grant in the form of subsidies from the federal budget of the Russian Federation under grants from the President of the Russian Federation for state support of young Russian scientists—PhDs— and leading scientific schools of the Russian Federation.

**Institutional Review Board Statement:** Not applicable.

**Informed Consent Statement:** Not applicable.

**Data Availability Statement:** Data sharing is not applicable to this article.

**Acknowledgments:** The authors are grateful to Ma A Chang for provided photos of black cardamom and Tam H. M. for botanical identification.

**Conflicts of Interest:** The authors declare no conflict of interest.

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
