# Peer review of "Improvement of In Vitro Seed Germination and Micropropagation of Amomum tsao-ko (Zingiberaceae Lindl.)"

_horticulturae, doi:10.3390/horticulturae8070640_

Round 1

Reviewer 1 Report

Review Horticulturae

Reviewer’s comments to manuscript horticulturae-1792373-peer-review-v2

1. General of reviewer´s comments

The authors Quyet V. Khuat et al. of the manuscript entitled “ Improvement of in vitro Seed Germination and Micropropagation of Black Cardamom (Amomum tsao-ko Crevost & Lemarié)made a significant effort to find the reason for the low germination of black cardamom seeds as well as to develop treatments that can improve germination and shoot multiplication in vitro. The results obtained provide important information on how to promote the propagation of this important spice and medicine plant cultivated in Asia. The results have scientific and practical value and worthy publication in the scientific journal Horiculturae. However, the main drawback of this paper is the lack of proper Discussion. Namely, since the authors previously performed similar experiments on germination and in vitro shoot multiplication of this plant (the results have already been published in related papers Khuat Van Quyet et al 2021 IOP Conf. Ser.: Earth Environ. Sci. 677 042065; Khuat Van Quyet et al 2021 IOP Conf. Ser.: Earth Environ. Sci. 848 012207), while the results of the present study were characterized as “improvements of the germination and micropropagation  . . ”, the authors should explain and discuss the presented results more in relation to the previously published ones, and not just list the plant species in which similar protocols have been applied. The authors must also explain what the improvements of the germination protocol are compared to the previously published results, because the same protocol (mechanical scarification) provided the highest germination rate in both studies. The authors did not explain why the previously used mechanical scarification gave 33% of seed germination compared to 68% in the present study? In addition, some calculation would be appropriate how improved germination can ensure more efficient planting of black cardamom.

On the other hand, in a previous paper by Khuat Van Quyet et al 2021 IOP Conf. Ser.: Earth Environ. Sci. 848 012207   the authors examined the effects of different cytokinins on the in vitro shoot multiplication of black caradamom and found that BAP at 1 mg/l was most favourable for shoot multiplication. In the present study, the authors listed BAP at 4 mg/l + 0.5 mg/l NAA as the most favourable combination for multiplication, shoot length and rooting, but some comparisons regarding plant quality are missing compared to plants regenerated on medium with lower BAP. Therefore, it would be necessary to provide some data on the quality of multiplied plants, especially since it is known that higher concentrations of cytokinins can often lead to vitrification, callusing, shortening of shoots and roots etc., so it is advisable to discuss this problem.

2. Abstract: Please rewrite to be more concise and to point out the most significant results of the study. It could be:

Black cardamom (Amomum tsao-ko Crevost & Lemarié) is a spice plant of great commercial value in Vietnam, but with limited propagation ability. Its seeds were characterised by a thick and hard seed coat, a small endosperm, and a small embryo which was the cause of the physical dormancy of the seeds and low germination. Attempts to improve the germination rate and achieve uniform germination included mechanical scarification, immersion in hot or cold water, acid scarification, and application of plant growth hormones. Although immersion of seeds in cold water or application of plant growth hormones (GA3 and NAA) showed positive effects on seed germination and subsequent seedling growth, mechanical scarification provided the highest rate of germination of black cardamom seeds (68.0%) and significantly shortened germination time (53.7 days) compared to control seeds (16.00% and 74.7days). On the other hand, an efficient micropropagation protocol has been established using shoot tip explants of in vitro grown seedlings. Murashige and Skoog (MS) medium supplemented with 4.0 mg/L benzylaminopurine (BAP) + 0.5 mg/L 1-naphtaleneaceticacid (NAA) proved to be most suitable for rapid multiplication and rooting, providing an average of 5.4 shoots per explant, 6.8 cm shoot length and16.2 roots per explant after 7 weeks of culture. Well-rooted black cardamom plantlets have been successfully adapted to ex vitro conditions. Bio-soil Fasco was more suitable for acclimatization with 48.9% survival rate, 23.3 cm plant length and 5.7 leaves per plant after 3 months of planting. Improved germination and multiplication protocols can be used to improve propagation performances and to develop elite of black cardamom planting material.

3. The Introduction is too long and contains a lot of information that is not relevant to the main point of the paper. I suggest to shorten a part of the description of the plant as well as the composition of black cardamom essential oil and to introduce data from recent literature and more relevant to the topic of the study e.g. seed priming and in vitro shoot multiplication of Zingiberaceae.

Please avoid duplication, as plant growing area (line 40 and line 55-57).

4. Also, avoid repeating the results in the Discussion as in lines 364-367. Discussion should focus more on the scientific explanation of the main results of the work.

Figure 5. Please add information on what the authors mean by T0, T1 . . .  etc., is very important for readers

Table 2. Please insert one column in front of the Treatment column with marked treatments T0-T10.

 Table 3. Please select, plant growth hormones or plant growth regulators and uniform throughout the manuscript.

5. If a Discussion is a separate chapter, pl avoid discussing the results in the Results chapter (lines 316-319).

6. Discussion needs to be reduced, rewritten according to the previous remarks. Avoid excessive repetition of results in the Discussion (lines 364-367).

7. Conclusion should be more concise emphasizing the main achievements of the study along with the future perspectives, avoid copying the sentences from the abstract and the main text in the Conclusion.

8. Too many references, especially the older ones. Please reduce the list of referencest. Replace older references with mostly recent ones eg. published in the last 5 years, which are more related to the topics of the study.

The paper is interesting and provides useful scientific and practical information related to improving the effects of seed priming and shoot multiplication in vitro on the propagation of black cardamom. However, due to the some drawbacks observed, I advise the authors to significantly improve the manuscript according to the given remarks. Since the manuscript cannot be considered for publication in Horticulturae in its current form, I recommend a major revision after the manuscript is considered for publication in Horticulturae.

Author Response

Dear Reviewer,

We want to thank the Reviewer, who have spent reading and additional commenting on our manuscript. We tried to significantly improve the manuscript after resubmitting. Corrections are made in the text and highlighted in color.

Point 1: Since the authors previously performed similar experiments on germination and in vitro shoot multiplication of this plant (the results have already been published in related papers Khuat Van Quyet et al 2021 IOP Conf. Ser.: Earth Environ. Sci. 677 042065; Khuat Van Quyet et al 2021 IOP Conf. Ser.: Earth Environ. Sci. 848 012207), while the results of the present study were characterized as “improvements of the germination and micropropagation  . . ”, the authors should explain and discuss the presented results more in relation to the previously published ones, and not just list the plant species in which similar protocols have been applied. The authors must also explain what the improvements of the germination protocol are compared to the previously published results, because the same protocol (mechanical scarification) provided the highest germination rate in both studies. The authors did not explain why the previously used mechanical scarification gave 33% of seed germination compared to 68% in the present study?

Response 1: We agree with the Reviewer's point of view and have added an explanation of what causes the difference in results between our two experiments in the "Discussion" section. As follows:

“The results of our previous study also gave similar results, but the germination rate of seeds was significantly improved in this experiment. The difference in the type of seeds used and the time of sowing in our two experiments is probably the cause for that difference. If in the first report we used dried seeds that were collected in October and dried in the open sun for 3 days, in this experiment we used fresh seeds that were collected in December. According to local people’s experience, the best time to sow black cardamom seeds is at the end of December and it must sow immediately the seed after harvest because it will lose its ability to germinate when stored for a long time. In a study on the effect of moisture in seeds on the germination of green cardamom seeds, Sangakkara U. R. (1990) found that reducing the moisture content of seeds significantly reduced and even lost the viability of them. Robert (1973) termed them recalcitrant seeds. Also bearing similar physiological characteristics to green cardamom seeds may be responsible for the difference in results in our two experiments. Further studies are needed to confirm this issue”.

Point 2: On the other hand, in a previous paper by Khuat Van Quyet et al 2021 IOP Conf. Ser.: Earth Environ. Sci. 848 012207 the authors examined the effects of different cytokinins on the in vitro shoot multiplication of black caradamom and found that BAP at 1 mg/l was most favourable for shoot multiplication. In the present study, the authors listed BAP at 4 mg/l + 0.5 mg/l NAA as the most favourable combination for multiplication, shoot length and rooting, but some comparisons regarding plant quality are missing compared to plants regenerated on medium with lower BAP. Therefore, it would be necessary to provide some data on the quality of multiplied plants, especially since it is known that higher concentrations of cytokinins can often lead to vitrification, callusing, shortening of shoots and roots etc., so it is advisable to discuss this problem.

Response 2: We agree with the Reviewer's point of view and have added the in vitro plantlet quality data obtained in the treatments (Table 3). On the other hand, the good shoot multiplication effect of MS medium supplemented with natural cytokinin (BAP) at high concentration and synthetic auxin (NAA) has been confirmed in some previous studies on some Zingiberaceous species such as green cardamom, ginger and turmeric. According to these studies, the in vitro plantlets obtained were of good quality. Similar to our study results on black cardamom.

Point 3. Abstract: Please rewrite to be more concise and to point out the most significant results of the study.

Response 3: We agree with the Reviewer's point of view and have rewritten the abstract according to the comments.

Point 4: The Introduction is too long and contains a lot of information that is not relevant to the main point of the paper. I suggest to shorten a part of the description of the plant as well as the composition of black cardamom essential oil and to introduce data from recent literature and more relevant to the topic of the study e.g. seed priming and in vitro shoot multiplication of Zingiberaceae.

Response 4: We agree with the Reviewer's point of view and have rewritten the introduction according to the comments.

Point 5:

Figure 5. Please add information on what the authors mean by T0, T1 . . .  etc., is very important for readers

Response: We agree and have added captions to the T0, T1 . . .  etc. on the figure.

Table 2. Please insert one column in front of the Treatment column with marked treatments T0-T10.

Response: In our opinion it is not necessary to add the column because there is already a caption for the figure.

Table 3. Please select, plant growth hormones or plant growth regulators and uniform throughout the manuscript

Response: We agree to replace and has rewritten in the revised manuscript.

Point 6. If a Discussion is a separate chapter, avoid discussing the results in the Results chapter (lines 316-319).

Response 6: We agree to replace and has rewritten in the revised manuscript.

Point 7. Discussion needs to be rewritten according to the previous remarks. Avoid excessive repetition of results in the Discussion (lines 364-367).

Response 7: We agree with the Reviewer's point of view and have rewritten the discussion according to the comments.

Point 8. Conclusion should be more concise emphasizing the main achievements of the study along with the future perspectives, avoid copying the sentences from the abstract and the main text in the Conclusion.

Response 8: We agree with the Reviewer's point of view and have rewritten the conclusion according to the comments.

Point 9. Too many references, especially the older ones. Please reduce the list of references. Replace older references with mostly recent ones e.g. published in the last 5 years, which are more related to the topics of the study.

Response 9: We agree with the Reviewer's point of view and have rewritten the references according to the comments.

We one more thanks the reviewer for detail analysis of our manuscript. We hope that the resubmitted version of the article has become better and more understandable to readers.

Sincerely,

Marat Khaliluev

Reviewer 2 Report

Comments and Suggestions for Authors

The manuscript submitted for review presents the results of the investigation of the effects of different seed treatment methods on the seed germination and seedling growth of black cardamom under in vitro conditions; determine the effect of different plant growth regulators (PGRs) on in vitro shoot multiplication and root formation of seedling shoot tips in order to develop a protocol that constant supply of viable and clean clonally propagated plant material for crop production. This study appears to be the first report of the morphological, anatomical, and germinating characteristics of black cardamom seeds collected in Vietnam. The Black cardamom is a spice plant with great commercial value in Vietnam and is used as a spice in many Asian countries, as flavoring agents and for pharmaceutical purposes by different industries. The area of black cardamom cultivation is constantly expanding. China is the major export market of Vietnamese cardamom. Annually, Lao Cai province exports to China ca. 4000 tons of dried capsule. In Vietnam, local people used mainly seeds and rhizome segments to cultivate black cardamom. Because plants grown from rhizome segments are susceptible to diseases caused by viruses, fungi, or bacteria, they often give lower yield and fruit quality than plants grown from seeds. Also, the destructive harvesting of the rhizomes for vegetative propagation seems not to be workable because there is always the possibility of losing the mother plant during this process. However, the practice of seed propagation of local people shows that cardamom seeds germinate slowly and unevenly, and seedlings are slow growth. Currently, the propagation method using plant tissue culture technology promises to produce many disease-free and uniform quality crops in a short time compared with traditional methods. Using disease-free plants obtained from in vitro propagation of black cardamom seedlings for production will solve the difficulties encountered by the two traditional propagation methods.

Studies in other species from Zingiberaceae family have shown that seed germination of these species was not fast and/or many seeds not germinate due to the presence of some kind of dormancy, possibly associated with its hard and impermeable nature of the seed coat. Also, in the above-mentioned species, various seed treatments (including mechanical scarification, soak in hot or cold water, acid scarification, and soak in plant growth hormones, mechanical scarification treatment to improve the germination rate of dried seeds) were applied to break the seed dormancy. Presently, similar studies in black cardamom are lacking.

In present study, seed coat ultrastructural features of the black cardamom using SEM were first reported; attempts to improve germination percentage and to attain uniform germination were made; an efficient micropropagation protocol using in vitro seedling shoot tips as explants was presented, which can be utilized for the development of elite planting material.

The manuscript is well structured and presented. The research methods applied are appropriate, comprehensive and sufficient to achieve the objectives of the study. The illustrative material (tables and figures) are representative and of good quality.

In order to improve the manuscript, I have some comments and suggestions:

Title:

In my opinion, it is not appropriate to be present in the title both, the common and scientific name of target species. I suggest the following title: Improvement of in vitro Seed Germination and Micropropagation of Amomum tsao-ko (Zingiberaceae Lindl.). The author name of the species (Crevost & Lemarié) is usually introduced in “Introduction”, and once introduced further it is not be displayed in the text. Also, in the text the species have to be cited as “A.tsao-ko”,or “black cardamom”, and not as it is cited - black cardamom (Amomum tsao-ko Crevost & Lemarié).

Keywords:

“black cardamom is more appropriate keyword instead “Amomum tsao-ko Crevost & Lemarié” which is present in the title

Introduction:

Line 83: In the sentence: “Studies in Alpinia malaccensis Roscoe [22], Alpinia galanga Willd. [23], …” it must to be added “other species from Zingiberaceae family”, and namely: : “Studies in other species from Zingiberaceae family as Alpinia malaccensis Roscoe [22], …”

Results:

3.1. Morphological and anatomical characteristics of black cardamom seeds

(1)               /lines 210-215/ Below is presented my suggestion for this part:

Testa (or seed coat): very hard, representing an effective barrier against water and air, and consisting of: Epidermis of testa (it is unnecessary) comprising one layer of cells, longitudinally elongated; Hypodermis  comprising one layer of cells, tangentially elongated; Below the hypodermis, a layer of large parenchymatous cells containing volatile oil; Below this layer is the pigment layer comprising several layers of brown cells; Endotesta comprising one layer of palisade sclerenchymatous cells, brown.

(3) Endosperm:

The sentence (lines 218-219): “The small endosperm, …. and the slow growth of seedlings.” is more suitable for the "Discussion" in the following form: “The established in the study morphological and anatomical characteristics of black cardamom seeds, as the very hard testa, small endosperm (i.e. low food storage in it ) and small embryo can be responsible for the difficulty of seed germination and the slow growth of seedlings.”

Discussion:

The first paragraph (lines 352-360) is more suitable for “Introduction”.

In conclusion, this manuscript is recommended for publication in “Horticulturae”, after consideration of the remarks shown.

Author Response

Dear Reviewer,

We want to thank the Reviewer, who have spent reading and additional commenting on our manuscript. We tried to significantly improve the manuscript after resubmitting. Corrections are made in the text and highlighted in color.

Point 1: Title: In my opinion, it is not appropriate to be present in the title both, the common and scientific name of target species. I suggest the following title: Improvement of in vitro Seed Germination and Micropropagation of Amomum tsao-ko (Zingiberaceae Lindl.). The author name of the species (Crevost & Lemarié) is usually introduced in “Introduction”, and once introduced further it is not be displayed in the text. Also, in the text the species have to be cited as “A.tsao-ko”, or “black cardamom”, and not as it is cited - black cardamom (Amomum tsao-ko Crevost & Lemarié).

Response 1: We agree to replace and has rewritten in the revised manuscript

Point 2: Keywords: “black cardamom” is more appropriate keyword instead “Amomum tsao-ko Crevost & Lemarié” which is present in the title

Response 2: We agree to replace and has rewritten in the revised manuscript.

Point 3: Introduction: Line 83: In the sentence: “Studies in Alpinia malaccensis Roscoe [22], Alpinia galanga Willd. [23], …” it must to be added “other species from Zingiberaceae family”, and namely: : “Studies in other species from Zingiberaceae family as Alpinia malaccensis Roscoe [22], …”

Response 3: We agree to replace and has rewritten in the revised manuscript.

Point 4: Results:

(1) /lines 210-215/ Below is presented my suggestion for this part: Testa (or seed coat): very hard, representing an effective barrier against water and air, and consisting of: Epidermis comprising one layer of cells, longitudinally elongated; Hypodermis comprising one layer of cells, tangentially elongated; Below the hypodermis, a layer of large parenchymatous cells containing volatile oil; Below this layer is the pigment layer comprising several layers of brown cells; Endotesta comprising one layer of palisade sclerenchymatous cells, brown.

(2) Endosperm: The sentence (lines 218-219): “The small endosperm,…. and the slow growth of seedlings.” is more suitable for the "Discussion" in the following form: “The established in the study morphological and anatomical characteristics of black cardamom seeds, as the very hard testa, small endosperm (i.e. low food storage in it) and small embryo can be responsible for the difficulty of seed germination and the slow growth of seedlings”.

Response 4: We agree to replace and has rewritten in the revised manuscript (lines 383-388).

Point 5. Discussion: The first paragraph (lines 352-360) is more suitable for “Introduction”.

Response 5: We agree to replace and has rewritten in the revised manuscript.

We one more thanks the reviewer for detail analysis of our manuscript. We hope that the resubmitted version of the article has become better and more understandable to readers.

Sincerely, Marat Khaliluev

Round 2

Reviewer 1 Report

Reviewer’s comments on the manuscript horticulturae-1792373-peer-review-v3

The authors have addressed all the objections suggested by the reviewer and made all the requested corrections so that the manuscript entitled   Improvement of in vitro Seed Germination and Micropropagation of Amomum tsao-ko (Zingiberaceae Lindl.)’   is significantly improved compared to the previous version. However, there are a few mistakes and ambiguous sentences, so I advise the authors to carefully check the language, preferably by a native English speaking person.

Some special points to note:

line 94   it should be  . . . ., and seedlings grow slowly

line 152 it should be . . . .  , and uniform quality crop plants in a short time . . .

line 417 , produced callus was (or produced calli were . . .) white . . .

lines 464-466 it could be    In black cardamom, the morphological and anatomical characteristics of the seeds found in the present study, such as the very hard testa, small endosperm (i.e. low food storage in it) and small embryo may be responsible for the difficulty of seed germination and slow growth of seedlings.

line 542 These authors instead of these workers

line 549 Please correct:  According TO our previous study . .....

line 552 “But in this study . . .” Please define, it is not clear to which study you are referring to

line 591 Pl explain and rewrite, the sentence is ambiguous“, , , ,the seed coat caused the major cause???

line 593 it could be  Immersion of seeds in cold water  . . . .

line 597 It is messy sentence, please rewrite to be more precise

lines 601-603 it could be: The results of the present study may be of great commercial importance as black cardamom is used as a spice in many Asian countries, as a flavouring agent and for pharmaceutical purposes in other industries.

Therefore, after the obligatory language check and correction of some errors, I recommend that this manuscript may be considered for publication in Horticulturae after minor revision.

Author Response

Dear Reviewer,

On behalf of ourselves and the co-authors, we thank you for your appreciation of our manuscript. We are confident that your comments and corrections will make our manuscript better. We made required changes in the manuscript and attempted to answer to all your questions.

Point 1: line 94   it should be  . . . ., and seedlings grow slowly

Response 1: We agree to replace and has rewritten in the revised manuscript.

Point 2: line 152 it should be . . . .  , and uniform quality crop plants in a short time . . .

Response 2: We agree to replace and has rewritten in the revised manuscript.

Point 3. line 417 , produced callus was (or produced calli were . . .) white . . .

Response 3: We agree to replace and has rewritten in the revised manuscript.

Point 4: lines 464-466 it could be: “In black cardamom, the morphological and anatomical characteristics of the seeds found in the present study, such as the very hard testa, small endosperm (i.e. low food storage in it) and small embryo may be responsible for the difficulty of seed germination and slow growth of seedlings.

Response 4: We agree to replace and has rewritten in the revised manuscript.

Point 5: line 542 These authors instead of these workers

Response: We agree to replace and has rewritten in the revised manuscript.

Point 6. line 549 Please correct:  According TO our previous study…

Response 6: We agree to replace and has rewritten in the revised manuscript.

Point 7. line 552 “But in this study . . .” Please define, it is not clear to which study you are referring to

Response 7: We rewritten it as “But in the current study…”

Point 8. line 591 Pl explain and rewrite, the sentence is ambiguous“, , , ,the seed coat caused the major cause???

Response 8: We rewritten it as “Seed germination of black cardamom wasn’t fast due to the associated with its hard and impermeable nature of the seed coat”.

Point 9. line 593 it could be  Immersion of seeds in cold water…

Response 9: We agree to replace and has rewritten in the revised manuscript.

Point 10. line 597 It is messy sentence, please rewrite to be more precise

Response 10: We agree to replace and has rewritten in the revised manuscript.

Point 11. lines 601-603 it could be: The results of the present study may be of great commercial importance as black cardamom is used as a spice in many Asian countries, as a flavouring agent and for pharmaceutical purposes in other industries.

Response 11: We agree to replace and has rewritten in the revised manuscript.

Additionally, we checked the entire manuscript for grammatical errors. We have made appropriate changes to the text of the manuscript.

We one more thanks the reviewer for detail analysis of our manuscript. You have greatly improved our manuscript with your valuable suggestions and recommendation.

Sincerely,

Marat Khaliluev